# 3D Spheroid Configurations Are Possible Indictors for Evaluating the Pathophysiology of Melanoma Cell Lines

**DOI:** 10.3390/cells12050759

**Published:** 2023-02-27

**Authors:** Hiroshi Ohguro, Megumi Watanabe, Tatsuya Sato, Fumihito Hikage, Masato Furuhashi, Masae Okura, Tokimasa Hida, Hisashi Uhara

**Affiliations:** 1Departments of Ophthalmology, Sapporo Medical University, S1W17, Chuo-ku, Sapporo 060-8556, Japan; 2Departments of Cardiovascular, Renal and Metabolic Medicine, Sapporo Medical University, S1W17, Chuo-ku, Sapporo 060-8556, Japan; 3Departments of Cellular Physiology and Signal Transduction, Sapporo Medical University, S1W17, Chuo-ku, Sapporo 060-8556, Japan; 4Departments of Dermatology, Sapporo Medical University, S1W17, Chuo-ku, Sapporo 060-8556, Japan

**Keywords:** 3D spheroid culture, melanoma, RNA sequencing, KRAS, SOX2

## Abstract

To study the molecular mechanisms responsible for inducing the spatial proliferation of malignant melanomas (MM), three-dimension (3D) spheroids were produced from several MM cell lines including SK-mel-24, MM418, A375, WM266-4, and SM2-1, and their 3D architectures and cellular metabolisms were evaluated by phase-contrast microscopy and Seahorse bio-analyzer, respectively. Several transformed horizontal configurations were observed within most of these 3D spheroids, and the degree of their deformity was increased in the order: WM266-4, SM2-1, A375, MM418, and SK-mel-24. An increased maximal respiration and a decreased glycolytic capacity were observed within the lesser deformed two MM cell lines, WM266-4 and SM2-1, as compared with the most deformed ones. Among these MM cell lines, two distinct cell lines, WM266-4 and SK-mel-24, whose 3D appearances were the closest and farthest, respectively, from being horizontally circular-shaped, were subjected to RNA sequence analyses. Bioinformatic analyses of the differentially expressed genes (DEGs) identified KRAS and SOX2 as potential master regulatory genes for inducing these diverse 3D configurations between WM266-4 and SK-mel-24. The knockdown of both factors altered the morphological and functional characteristics of the SK-mel-24 cells, and in fact, their horizontal deformity was significantly reduced. A qPCR analysis indicated that the levels of several oncogenic signaling related factors, including KRAS and SOX2, PCG1α, extracellular matrixes (ECMs), and ZO1 had fluctuated among the five MM cell lines. In addition, and quite interestingly, the dabrafenib and trametinib resistant A375 (A375DT) cells formed globe shaped 3D spheroids and showed different profiles in cellular metabolism while the mRNA expression of these molecules that were tested as above were different compared with A375 cells. These current findings suggest that 3D spheroid configuration has the potential for serving as an indicator of the pathophysiological activities associated with MM.

## 1. Introduction

A malignant melanoma (MM) originates from melanocytes, in which a photoprotective pigment, melanin is synthesized, and thus can be found in various organs that contain cells that produce this pigment including the eye, the gastrointestinal tract, genitalia, sinuses, and meninges, in addition to the skin [1]. The most common risk for developing MM is ultraviolet (UV) injury to the skin and the incidence of such occurrences are increasing worldwide [2,3]. Recent advances in the treatment of MM over the last 10 years have resulted in remarkable improvements in the survival and quality of life, even in patients with advanced stages, based upon new MM therapies using targeted inhibitors of BRAFV600E, such as dabrafenib, and MEK kinases, such as trametinib, that are involved in the mitogen-activated protein kinase (MAPK) pathway in conjugation with immune-based strategies [4,5,6,7]. In fact, approximately 60% of the objective response rate was detected in patients who had been treated with a combination of ipilimumab (anti-CTLA-4) and nivolumab (anti-PD-1) [8] resulting in 52% of these patients being alive at 5 years after receiving this combination as compared with 26% of those treated with ipilimumab alone [9]. However, in contrast to such dramatic progress, currently available therapies do not always result in patients with metastatic MM being cured, particularly in patients with tumors that lack a BRAF mutation as well as in patients with rare MM subtypes, such as uveal, acral, and mucosal MM due to either primary or acquired resistance following an initial response toward these targeted and non-targeted immune-based strategies [10].

During the development of translation research related to MM, several disease models have been used to obtain fundamental discoveries in MM biology and these studies have resulted in the development of new therapeutic targets and the identification of several clinical markers. For example, several studies using animal models have revealed the efficacy of using anti-PD-1 and anti-CTLA-4 antibodies to inhibit tumor immunity [11,12,13], leading to therapy for patients with MM and other malignancies [14,15]. As an alternative approach, in vitro three-dimensional (3D) culture models, which have been used in studies concerning the structure and molecular and cellular mechanisms of tumors as compared to the conventional two-dimensional (2D) models, have also been utilized in research related to malignant tumors including MM [16]. In fact, several studies reported that these in vitro 3D tumor models facilitate a better understanding of cell to cell and cell to matrix interactions since they replicate the architecture and cellular heterogeneity of tumor tissues more closely [17,18]. Thus, consequently, these in vitro 3D models are now frequently used for the screening of new antitumor drugs [19,20]. Among the several in vitro 3D models, 3D spheroids, which are simpler forms and can be produced in a nonadherent surface manner, have been the most widely used [21,22], and they have emerged as a useful tool for modeling a number of human diseases, in addition to malignant tumors including MM [23,24,25]. Interestingly, using a 3D MM spheroid model, Ahmed and Haass reported that the proliferative and invasive efficacies could be defined by the high and low expression, respectively, of the microphthalmia-associated transcription factor (MITF) [26].

Independently, to establish in vitro models that replicate several disease states, by using a 3D drop cell culture method, we recently succeeded in producing 3D spheroids of several non-cancerous cells including 3T3-L1 preadipocytes, human orbital fibroblasts (HOF) [27], human trabecular meshwork (HTM) cells [28], human conjunctival fibroblasts (HconF) [29], as well as others. Concerning functional aspects of the 3D spheroids, our studies relating to 3T3-L1 cells identified significant differences in biological activities between the 3D spheroids and conventional 2D cultured cells. The 3D 3T3-L1 spheroids were found to undergo adipogenic differentiation much more efficiently as compared to the 2D cells [30]. Quite interestingly, regarding the 3D spheroid configurations, although the obtained 3D spheroids originated from non-cancerous cells and were all shown to be globe-shaped [27,28,29], non-globe shaped 3D spheroids were obtained from cancerous cells (personal communication). In fact, it was reported that such non-globe shaped configurations were also detected in some of the 3D MM spheroids [31]. Therefore, this collective evidence prompted us to hypothesize that 3D spheroid configurations may be valuable indicators for malignancy as well as other biological aspects. However, no study focusing on the underlying mechanisms responsible for inducing 3D spheroid architectures, that is, globe shaped or non-globe shaped, has been reported so far.

Therefore, to study this issue, to elucidate the unidentified molecular mechanisms responsible for inducing the 3D spheroid architecture, we prepared 3D spheroids from five different MM cell lines (SK-mel-24, MM418, A375, WM266-4, and SM2-1) and dabrafenib and trametinib resistant A375 (A375DT) by a 3D drop cell culture method and the resulting spheroids were subjected to real time cellular metabolism analysis to evaluate the biological activities. In addition, among these five cell lines, we selected two distinct MM cells, one of which was horizontally circular shaped and the other non-horizontally circular shaped and subjected them to RNA sequence analysis to elucidate possible master regulatory genes responsible for these structural diversities.

## 2. Materials and Methods

### 2.1. 2D Planar and 3D Spheroid Cultures of MM Cell Lines

Five human MM cell lines including (1) A375 (CRL-1619™, American Type Culture Collection, Manassas, VA, USA), (2) SK-mel-24 (HTB-71, American Type Culture Collection, Manassas, VA, USA), (3) WM266-4 (CVCL_2765, American Type Culture Collection, Manassas, VA, USA), (4) SM2-1 (acral lentiginous MM obtained from Dr. Hiroshi Murata, Shinshu University School of Medicine; Matsumoto, Nagano, Japan) [32], and (5) MM418 (a generous gift from Dr. Peter Parsons, QIMR, Brisbane, Australia) that were 2D and 3D cultured by methods described in previous reports using 3T3-L1 preadipocytes and human orbital fibroblasts [27,30,33,34] (Information concerning methods used in this study are shown in Appendix A). At Day 7, both the 2D and 3D cultured cells each collected were further processed for use in the analyses described below.

A375DT (A375-dabrafenib/trametinib resistant) cell line was established from A375 cells by chronic treatment with gradually increasing concentrations of dabrafenib and trametinib, as described previously [35], and were selected for monoclones using the limiting dilution method.

### 2.2. Measurement of Mitochondrial and Glycolytic Functions of Various MM Cell Lines

Oxygen consumption rate (OCR) and extracellular acidification rate (ECAR) of the 2D cultured MM cell lines including SK-mel-24, SK-mel-23, MM418, A375, WM266-4, and SM2-1 were measured using a Seahorse XFe96 real-time metabolic analyzer (Agilent Technologies, Santa Clara, CA, USA) according to the manufacturer’s instructions (Information concerning methods used in this study is shown in Appendix A).

### 2.3. Phase Contrast Microscopy of the 3D Spheroids Derived from Various Human MM Cell Lines

The morphology of the 3D MM spheroids was observed by a phase-contrast microscope (Nikon ECLIPSE TS2; Tokyo, Japan) and a micro-monitoring camera equipped with a micro-squeezer (MicroSquisher, CellScale, Waterloo, ON, Canada) as described previously [27].

### 2.4. Analyses of RNA Sequence Gene Function and Metabolic Pathways

Total RNA isolation was performed from 2D confluent cells ofWM266-4 and SK-mel-24 in a 150 mm dish as above (n = 3) using a RNeasy mini kit (Qiagen, Valencia, CA, USA) according to the manufacturer’s instructions. RNA content and quality were measured using a NanoPhotometer^®^ P330 (IMPLEN, Los Angeles, CA, USA) and an Agilent 2100 Bioanalyzer (Agilent Technologies, Massy, France), respectively. As RNA quality was suitable for RNA sequencing, and quantitative real time PCR, the samples with an RNA integrity number (RIN) > 8.5 were confirmed in advance. The total RNA was depleted of ribosomal RNA using NEBNext^®^ Poly(A) mRNA Magnetic Isolation Module (Cat. # E7490, New England BioLabs, Ipswich, MA, USA). The rRNA-depleted RNA was processed according to the manufacturer’s protocol to convert to cDNA using a TruSeq RNA Sample Preparation Kit (Illumina, San Diego, CA, USA) and final sequence-ready libraries with the NEBNext Ultra II RNA library prep kit (Cat. #E7760, New England BioLabs, Ipswich, MA, USA). Then their quality and quantity were determined using an Agilent 2100 Bioanalyzer and KAPA Library Quantification Kit (KAPA Biosystems, Wilmington, MA, USA), respectively. Thereafter, they were subjected to NovaSeq 6000 and GenoLab M sequencing in PE150 mode. Sequence data were filtered by removal of the adapter sequence, ambiguous nucleotides, and low-quality sequences using FastQC (version 0.11.7) and Trimmomatic (version 0.38) software. Then, these clean reads were mapped to the reference genome sequence with a perfect match or one mismatch method via HISAT2 tools software [36]. The corresponding genome references were downloaded from ensemble database by https://ftp.ensembl.org/pub/release-101/fasta/homo_sapiens/dna, accessed on 5 September 2022. The read counts for each respective gene and statistical analysis of the differentially expressed genes were calculated by featureCounts (version 1.6.3) and DESeq2 (version 1.24.0), respectively. Statistical significance was determined by the empirical analysis, and genes with fold-change ≥ 2.0 and FDR adjusted *p* < 0.05 and q < 0.2 were assigned as differentially expressed genes (DEG).

A gene ontology (GO) enrichment analysis [37] as well as an ingenuity pathway analysis (IPA, Qiagen, https://www.qiagenbioinformatics.com/products/ingenuity-pathway-analysis, accessed on 5 September 2022) [38] were performed to estimate gene function, as described in a recent report [33].

### 2.5. Transfection of siRNA

For the knockdown of KRAS and/or SOX2 of SK-mel-24 cells, 2D cultured cells were incubated with Lipofectamine 3000 (Invitrogen, Carlsbad, CA, USA) containing either 30 nM of (1) siRNA Universal Negative Control (#SIC001, Mission siRNAs, Sigma Aldrich, St. Louis, MO, USA), (2) KRAS siRNA-1 (#SASI_Hs01_00202556, Mission siRNAs, Sigma Aldrich), (3) KRAS siRNA-2 (#SASI_Hs01_00202557, Mission siRNAs, Sigma Aldrich), (4) SOX2 siRNA-1 (#SASI_Hs01_00050572, Mission siRNAs, Sigma Aldrich), (5) SOX2 siRNA-1 (#SASI_Hs01_00050573, Mission siRNAs, Sigma Aldrich) or (6) both KRAS siRNA-1 and SOX2 siRNA-2 for 48 h. according to the manufacturer’s protocol. After replacing the siRNA-Lipofectamine 3000 complexes media with normal growth medium to reduce toxicity, the cells were subjected to real time cellular metabolic analyses using a Seahorse XFe96 real-time metabolic analyzer (Agilent Technologies, Santa Clara, CA, USA), or a further 3D spheroid culture, as above.

### 2.6. Other Analytical Methods

Total RNA extraction, reverse transcription, real-time PCR, and the quantification of the respective genes using specific primers (Appendix A) were described in a previous report [30]. All statistical analyses were performed using the Graph Pad Prism 8 (GraphPad Software, San Diego, CA, USA). The statistical differences among groups were determined using a Student’s t-test for two group comparison or two-ANOVA followed by a Tukey’s multiple comparison test. Data are expressed as the arithmetic mean ± the standard error of the mean (SEM).

## 3. Results

To establish 3D human MM spheroids, human MM cell lines including WM266-4, SM2-1, A375, MM418, and SK-mel-24, were subjected to a 3D drop culture by which we recently succeeded in producing globe-shaped 3D spheroids of several non-cancerous cells including 3T3-L1 preadipocytes [30], human orbital fibroblasts (HOF) [27], Graves’ disease related HOF (GHOF) [27], human trabecular meshwork (HTM) [28], and human conjunctival fibroblasts (HCF) [29]. As shown in Figure 1, we were also able to obtain 3D spheroids from all five human MM cell lines. However, and quite interestingly, the shapes of those 3D MM spheroids (Figure 1A,B) were deformed and quite different from the globe-shaped 3D spheroids that were obtained from non-cancerous cells as above (Figure 1C). In addition, a significant variation was observed among the cell lines. To determine the degree of deformity, the ratios of the outer circle and the inner circle of the horizontal view of the PC image of the 3D spheroid were measured and the resulting values were compared among the MM cells (Figure 2). Among these, the horizontal configuration of the 3D spheroids derived from WM266-4 was the closest to the non-cancerous globe-shape 3D spheroids, and the degree of deformity of these spheroids was increased in the order of WM266-4, SM2-1, A375, MM418, and SK-mel-24. Alternatively, at a vertical view of the PC images, we observed disc-shaped configurations with some variations among the five MM cell lines.

To elucidate functional aspects related to these five MM cell lines, as their cellular mitochondrial and glycolytic function, oxygen consumption rate (OCR) and extracellular acidification rate (ECAR) were simultaneously monitored in real-time to estimate mitochondrial respiration and glycolytic ability by using a Seahorse XFe96 Bioanalyzer. As shown in Figure 3, the changes in OCR in response to an FCCP injection, which reflects maximal mitochondrial respiratory capacity (panel C), in response to an oligomycin injection, which reflects coupling efficiency (panel D), and the changes of ECAR in response to oligomycin injection, which reflects glycolytic reserve (panel E), were significantly different among these five MM cell lines. Interestingly, relatively higher maximal respiration (panel C) and lower glycolytic reserve (panel E) were observed within the two MM cell lines that were closer to having a horizontally circular-shape, WM266-4 and SM2-1, as compared with the other more deformed three cell lines. In addition, coupling efficiency, the proportion of oxygen consumed for ATP synthesis compared to that for proton leakage, was also higher in WM266-4, SM2-1, and A375 as compared to the others (panel D). These data suggest that the morphological characteristics of MM cells cultured under 3D conditions are closely related to their metabolic characteristics, and that mitochondrial respiratory capacities tend to be higher and glycolytic capacities tend to be lower in cells of spheroids that are horizontally circular shaped compared to those in non-horizontally circular shaped spheroids.

To investigate the unidentified and underlying mechanisms responsible for inducing such a horizontal deformity in the 3D spheroid configuration among these human MM cell lines, the two most contrasting cell lines, MW266-4 and SK-mel-24, in which the deformity was minimal and maximal, respectively, were subjected to an RNA sequence analysis. As shown in the heat map (Figure 4A) and MA and volcano plots (Figure 4B,C), the significantly up-regulated 2880 and down-regulated 8756 differentially expressed genes (DEGs, a significance level of <0.05 (FDR) and an absolute fold-change of ≥2) of MW266-4 as compared to SK-mel-24 were identified (a list of the up-regulated and down-regulated genes is shown in Appendix A).

To elucidate the possible functional roles of the above DEGs that were detected, a GO enrichment analysis and an ingenuity pathway analysis (IPA) (Qiagen, Redwood City, CA) were employed. Among the categories of diseases and disorders, and molecular and cellular functions, the IPA analysis indicated that cancer and several cellular functions including organismal injury and abnormalities, endocrine system disorders, gastrointestinal diseases, and neurological diseases were most likely involved, based upon obtained DEGs (Appendix A). In addition, an IPA analysis narrowed the cancer related categories into melanoma as the highest candidate of the DEG related disease (Table 1). This observation rationally confirms that the present RNA sequence analysis is acceptable for our research purposes.

To elucidate which regulatory mechanisms are involved within the diversity between 3D architectures of both MM cell lines, a GO analysis and IPA up-stream and causal network analyses were performed. The results of the GO analysis based on up-regulated and down-regulated DEG suggest that several cellular mechanisms may be related, as shown in Appendix A. Furthermore, as shown in Table 2 and Table 3, nine and five candidate genes as possible up-stream regulators and master regulators for the causal networks were detected. Quite interestingly, among these, KRAS and SOX2 (Appendix A) were identified as common factors, suggesting that both factors may be critical master factors that are involved in regulating the diversity of the 3D architectures among these five MM cell lines. In fact, it was recently reported that SOX2 and KRAS have interconnections [39,40], and that KRAS is mutated in 1.3% of skin cancers [39]. To verify this possibility, we next examined whether siRNA-induced knockdown of KRAS and/or SOX2 would impact the cellular metabolic functions of SK-mel-24 cell, and their 3D spheroid shape which is the most deformed among the five MM cells. As shown in Figure 5A, real time cellular metabolic analyses demonstrated that KRAS knockdown significantly suppressed abnormally enhanced glycolytic capacity of SK-mel-24 as well as mitochondrial maximal respiration and coupling efficiency. In contrast, the effect of SOX2 knockdown on metabolic properties was only pronounced in mitochondrial maximal respiration. Such effects were similarly observed with two different siRNAs for KRAS and SOX2 (Appendix A). Interestingly, the knockdown of either KRAS or SOX2 significantly reduced horizontal deformation and increased the size of 3D SK-mel-24 spheroid, while the double-knockdown of KRAS and SOX2 did not induce the enlargement of the 3D spheroid. Taken together, these collective results indicate that KRAS and SOX2 were at least involved in the deformation of the 3D SK-mel-24 spheroids, but that their roles for metabolic functions were only partial. Therefore, additional factors other than KRAS and SOX2 appears to be involved in the difference in the morphology and the cellular metabolic functions among the five MM cell lines.

To investigate this further, mRNA expression analyses of several possible linked molecules including (1) oncogenic signaling related factors (KRAS, SOX2, MITF, BRAF, FOS, and STAT3), (2) PCG1α, and (3) ECMs (COL4, COL6, FN, and αSMA) and tight-junction (TJ) related molecule (ZO-1) were carried out among 2D and 3D cultured five MM cell lines (Table 4 and Appendix A). Compared to WM266-4, whose 3D spheroid was closest to the horizontally circular-shape among the five MM cell lines, the mRNA expressions of the other four 2D and 3D cultured MM cell lines were similarly altered as follows; (1) oncogenic signaling related factors; *KRAS* (up-regulated except 2D A375), *SOX2* (down-regulated except 3D SM2-1), *STAT3* (down-regulated; 2D SM-2-1 and 2D and 3D A375; up-regulated; MM418 and SK-mel-24), *BRAF* (up-regulated; MM418 and SK-mel-24, down-regulated; 3D SM2-1), *FOS* (up-regulated; 2D MM418, down-regulated; 3D A375 and SK-mel-24), *MITF* (down-regulated; except SM2-1 and 3D MM418), (2) *PCG1α* (down-regulated), and (3) ECMs and TJ related molecule; *COL1* (down-regulated except for up-regulated 2D SK-mel-24), *COL6* (down-regulated except 3D A375), *FN* (down-regulated except for 3D SM-21 and up-regulated 2D MM418), *aSMA* (down-regulated except 3D MM418) and *ZO1* (down-regulated except A375). Therefore, based upon these observations, we speculate that in addition to KRAS and SOX2, several additional oncogenic signaling related factors, namely, PCG1α, ECMs, and TJ related molecules may be related to the unidentified mechanisms for inducing differences in the horizontally deformed 3D MM spheroids. In addition, since these oncogenic signaling related factors that were evaluated as above are closely related to the pathogenesis of MM, such morphology-dependent alteration of their gene expression rationally suggests that anti-MM drugs may also affect 3D MM spheroid morphology.

Among these MM cell lines, it is known that A375 is an MM cell line harboring BRAF V600E mutations which leads to the activation of the RAF/MEK/ERK mitogen-activated protein kinase (MAPK) cascade regulating cellular proliferation, differentiation, and survival signaling [6,41]. Based upon these findings, BRAF inhibitors (BRAFi) such as dabrafenib [42], the MEK inhibitors (MEKi) such as trametinib [43], and combinations thereof [44] that have recently been used for treating melanoma resulted in significantly improved overall survival in metastatic melanoma patients with BRAFV600E mutations. However, such clinical benefits are usually transient due to the rapid emergence of acquired resistance in most patients. In fact, such acquired resistance against BRAFi and/or MEKi has been replicated in human MM cell lines such as A375 MM cells [35,45,46]. Using this information, we next investigated the effects of the acquired drug resistance of BRAFi and/or MEKi on 3D spheroid configuration, cellular metabolic functions, and the mRNA expression of several factors as above for A375MM cells. Quite surprisingly, as compared with A375, A375DT showed globe shaped 3D spheroid that resembled those obtained from non-cancerous cells, as shown in Figure 1 panel C, and remarkable increases in maximal mitochondrial respiration and glycolytic capacity, suggesting an energetic alteration in cellular metabolism in A375DT (Figure 6). Furthermore, the mRNA expression of A375DT of most of the oncogenic signaling related factors and other molecules as above were all upregulated as compared with A375, except for *PCG1α* (3D), *COL4* (3D), *FN* (2D), and *αSMA* (2D and 3D) (Table 5 and Appendix A).

Therefore, these collective data reported herein suggest that it is possible that the critical regulatory factors KRAS and SOX2, may significantly be influenced, not only by the 3D architectures of MM cell lines, but also by their physiological and pathological characteristics with several oncogenic signaling related factors as well as cell architecture supporting molecules.

## 4. Discussion

Although several biological aspects of 3D cultures of MM cell lines have been extensively characterized and are now used as in vitro models for studying the pathological and therapeutic significance of MM [47,48], studies focusing on their 3D spheroid configurations remain quite limited. The findings reported in this study indicate that 3D spheroid configurations are exclusively distinct among cell lines, even though all were categorized as MM related ones. Quite interestingly, the cellular metabolic activities of mitochondria and glycolysis fluctuated in parallel with the degree of transformed horizontal configurations of the 3D MM spheroids. Furthermore, RNA sequence and bio-informative analyses revealed that KRAS and SOX2 represent possible master regulatory genes that are involved in this type of 3D spheroid architecture. In addition, a qPCR analysis indicated that several factors that are related to oncogenic signaling, including KRAS and SOX2, PCG1α, extracellular matrixes (ECMs) and ZO1 were significantly altered among these five MM cell lines. Furthermore, unlike A375, the dabrafenib and trametinib resistant A375 (A375DT) cells formed globe shape 3D spheroids with a more energetic metabolic state and the profiles for the mRNA expressions of these molecules were different. Therefore, based on these results, we speculated that the 3D spheroid configuration may be related to the pathophysiological activities of MM. In fact, in our recent studies, we found that the biological activities of 3T3-L1 preadipocytes were significantly different between their 2D and 3D cell cultures, that is, the efficiencies of adipogenesis and the expression levels of adipogenesis related factors and several ECM proteins were substantially stimulated in the 3D 3T3-L1 spheroids compared to 2D cultures of 3D 3T3-L1 cells [30,49].

As possible mechanisms responsible for the diversity between the 2D and 3D cultures of 3T3-L1 preadipocytes, our recently reported transcriptome analysis using an IPA upstream analysis identified STAT3 as the master upstream gene as the regulator responsible for inducing the diverse biological properties between these culture systems [49]. Among the STAT family proteins, STAT1 to STAT6 [50], it was revealed that STAT3 functionally plays an important role in manipulating the biological activities of cancer cells, by affecting their energy metabolism, that is, the metabolism of glucose and lipids [51,52]. Furthermore, since STAT3 is also involved in gravity-induced biological activities [53,54], these observations strongly suggest that such biological difference between 2D and 3D culture systems are caused by STAT3 related signaling mechanisms, suggesting that similar mechanisms may also be involved in the diversity observed within the 3D spheroid configurations of MM cell lines. In fact, in the current study, the gene expression of STAT3 as well as BRAF indeed fluctuated to reflect the 3D architectures of the 3D spheroids among the five MM cells. Interestingly, findings reported in recent studies suggest that STAT3 signaling is closely linked with KRAS and SOX2 related networks as follows; (1) STAT3 regulates the epithelial differentiation of malignant tumors caused by oncogenic KRAS, and thus, STAT3 acts as a regulator for cellular plasticity and the inhibition of the epithelial mesenchymal transition (EMT) linked with metastasis [55], (2) STAT3 can be upregulated in cancer stem cells [56] together with SOX2 in clustered circulating tumor cells, resulting in a higher potential for developing metastasis [57], and (3) a BRAF inhibitor initiates the STAT3 activation causing the up-regulation of SOX2 and CD24 resulting in an increased tolerance against BRAF inhibitors [58,59]. Therefore, these collective findings strongly suggest that STAT3 related SOX2 and KRAS networks may be involved in tumor metastasis and drug resistance in MM cells, which prompted us to speculate that the 3D spheroid configurations of MM cells may be valuable indicators for estimating their pathological activities. In fact, to effectively target oncogenic RAS including KRAS with their downstream signaling and metabolic pathways during tumorigenesis, we examined targets that were connected with several other oncogenic driver genes including SOX2 [39,40]. Furthermore, it was shown that SOX2 modulates the levels of MITF, a key determinant of the MM phenotype [60], in human melanocytes, and MM lines in vitro [61]. In fact, in the current study, a significant down-regulation or up-regulation in the mRNA expressions of MITF was observed within A375, MM418, and Sk-mel-24, as compared with WM-266-4 and of A375DT compared with A375, respectively. In addition, the mRNA expression of ECMs, which are regulated by several transcription factors including KRAS [62], SOX2 [63], STAT3 [64], and MITF [65], were also modulated by the 3D spheroid morphology as well as drug resistance.

However, as of writing, this conclusion remains speculative and the following study limitations would need to be investigated regarding these interesting and unidentified issues. These issues also include the relationship between horizontal and vertical deformity of the 3D MM spheroid and cellular physiology, the degree of malignancy, drug resistance, and others. Thus, to understand the biological correlations between 3D spheroid configurations and the tumor pathogenesis of MM in more detail, additional investigations with the objective of identifying additional unidentified factors related to KRAS and SOX2 will be our next projects.

## Figures and Tables

**Figure 1 cells-12-00759-f001:**
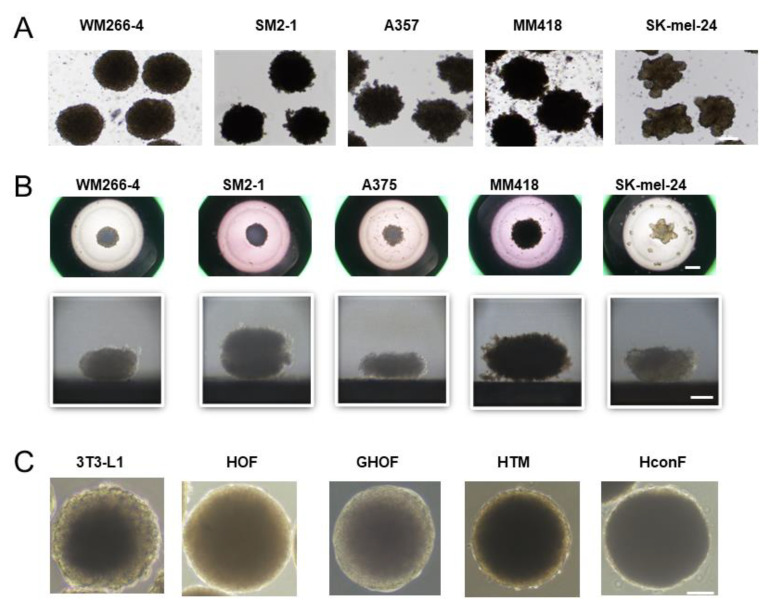
Representative PC (phase contrast microscopy) images of 3D spheroids of 5 malignant melanoma (MM) cell lines and 5 non-cancerous cells. Panel (**A**); representative PC images of the 3D spheroids obtained from 5 MM cell lines; WM266-4, SM2-1, A375, MM418, and SK-mel-24. Panel (**B**); the downward (upper) and lateral (lower) images of each single 3D spheroid from 5 MM cell lines; WM266-4, SM2-1, A375, MM418, and SK-mel-24. Panel (**C**); representative phase contrast images of collected 3D spheroids produced from non-cancerous cells including 3T3-L1 cells, human orbital fibroblasts (HOF), Graves diseases related HOF (GHOF), human trabecular meshwork (HTM) cell, human conjunctival fibroblasts (HconF). Scale bar; 100 μm.

**Figure 2 cells-12-00759-f002:**
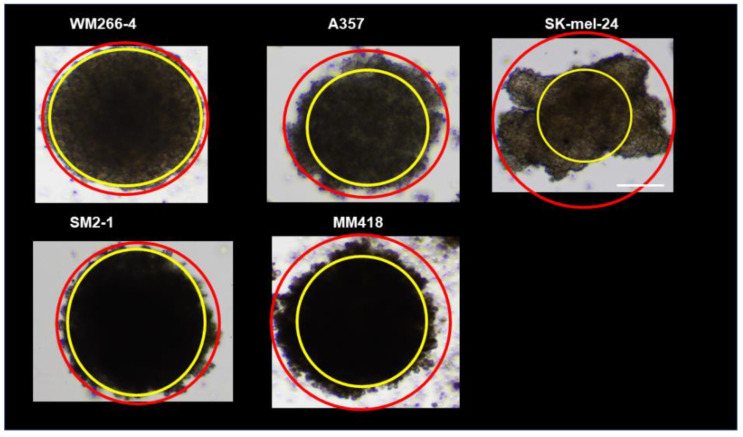
Eccentricity measurement of five MM cell lines. To estimate degrees of the deformity of each of the MM cell lines, horizontal images were obtained and their eccentricity rates (%), inner ring area (yellow)/outer ring area (red), were calculated. For example, WM266-4 (83.7%), SM2-1 (78.2%), A375 (58.7%), MM418 (55.5%), and SK-mel-24 (27.8%). Scale bar; 100 μm.

**Figure 3 cells-12-00759-f003:**
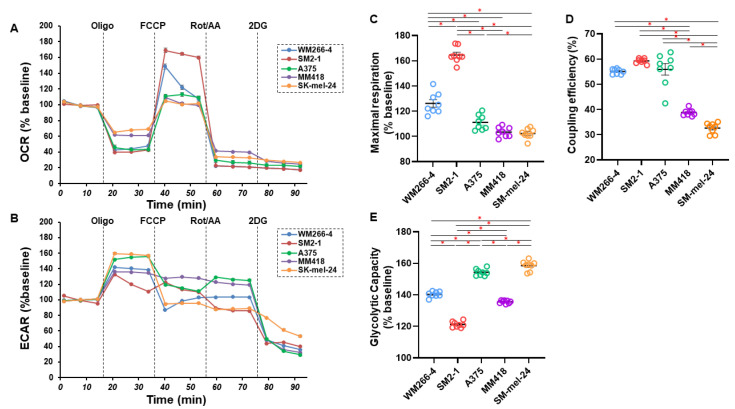
Measurement of the mitochondrial and glycolytic functions of five MM cell lines. The mitochondrial and glycolysis function of the five different 2D cultured MM cell lines; WM266-4, SM2-1, A375, MM418, and SK-mel-24 were measured using a Seahorse XFe96 Bioanalyzer. The oxygen consumption rate (OCR) and extracellular acidification rate (ECAR) before drug injection (at the baseline) were determined as 100%. The fluctuations were then sequentially monitored by the following injections: (i) oligomycin (a complex V inhibitor), (ii) FCCP (a protonphore), (iii) rotenone/antimycin (complex I/III inhibitors), and (iv) 2-DG (a hexokinase inhibitor) (OCR; panel **A** and ECAR; panel **B**). WM266-4 and SM2-1 demonstrated (1) a relative higher maximal respiration, defined as the difference between an average of OCR in the presence FCCP and an average of OCR at baseline were relatively higher as compared with others (panel **C**), and (2) a low glycolytic capacity, defined as the difference between the average of ECAR in the presence of oligomycin and the average of ECAR at the baseline (panel **E**). WM266-4, SM2-1 and A375 showed (3) a higher coupling efficiency, defined as the percentage of total baseline respiration that is sensitive to oligomycin (panel **D**). Indices of maximal respiration and glycolytic capacity are expressed as % baseline. * *p* < 0.05.

**Figure 4 cells-12-00759-f004:**
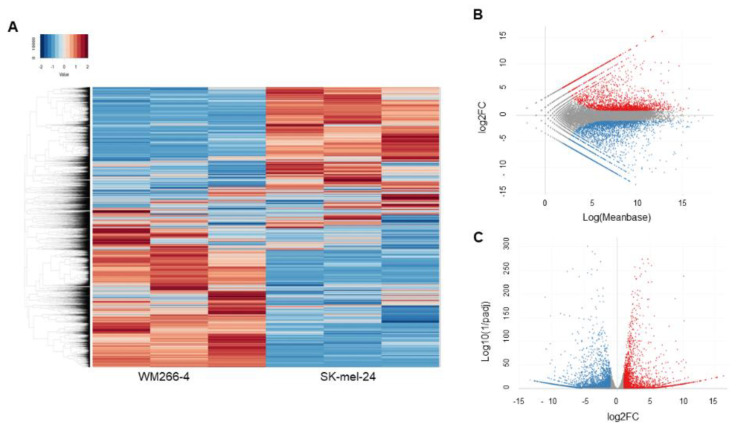
Differently expressed genes (DEGs) between WM266-4 and SK-mel-24. DEGs (cutoff false discovery rate (FDR) < 0.05 and/or the magnitude of change ≥ 2) between 2D cultured WM266-4 and SK-mel-24 cells were demonstrated by hierarchical clustering heatmaps (panel **A**), M-A plots (panel **B**) and volcano plots (panel **C**). Colored bars and points represent DEGs that are either over-expressed (red) or under-expressed (blue) in WM266-4 with SK-mel-24.

**Figure 5 cells-12-00759-f005:**
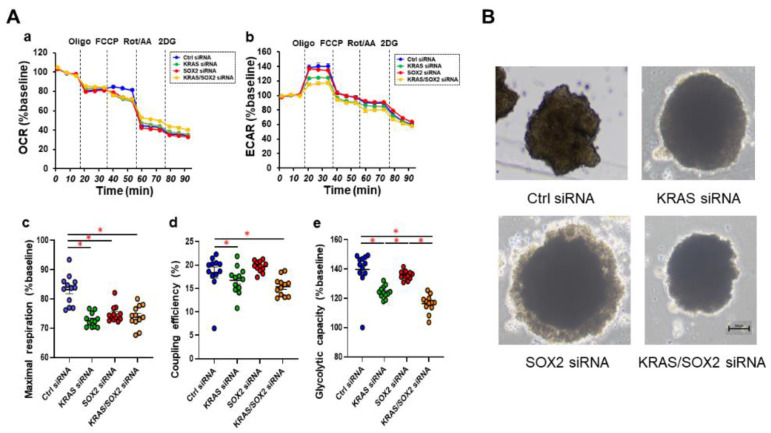
Effects of siRNA-mediated KRAS and/or SOX2 suppressions on the mitochondrial and glycolytic functions of 2D cultured SK-mel-24 cells and the 3D SK-mel-24 spheroid morphology. Effects of KRAS siRNA-1 and/or SOX2 siRNA-1 on the mitochondrial and glycolysis function of the 2D cultured SK-mel-24 cells and the horizontal morphology of the 3D SK-mel-24 spheroids were determined using a Seahorse XFe96 Bioanalyzer and a phase-contrast microscopy, respectively. Panel (**A**); the oxygen consumption rate (OCR) and extracellular acidification rate (ECAR) before drug injection (at the baseline) were determined as 100%. The fluctuations were then sequentially monitored after the following injections: (i) oligomycin (a complex V inhibitor), (ii) FCCP (a protonphore), (iii) rotenone/antimycin (complex I/III inhibitors), and (iv) 2-DG (a hexokinase inhibitor) (OCR; panel **a**, and ECAR; panel **b**). Maximal respiration (panel **c**) was defined as the difference between the average of OCR in the presence FCCP. Coupling efficiency (panel **d**) was defined as the percentage of total baseline respiration that is sensitive to oligomycin. Glycolytic capacity (panel **e**) was defined as the difference between the average of ECAR in the presence of oligomycin and the average of ECAR at the baseline. Indices of mitochondrial respiration and glycolytic capacity are expressed as % baseline. * *p* < 0.05. Panel (**B**); representative PC images of their 3D spheroids are shown (scale bar; 100 μm).

**Figure 6 cells-12-00759-f006:**
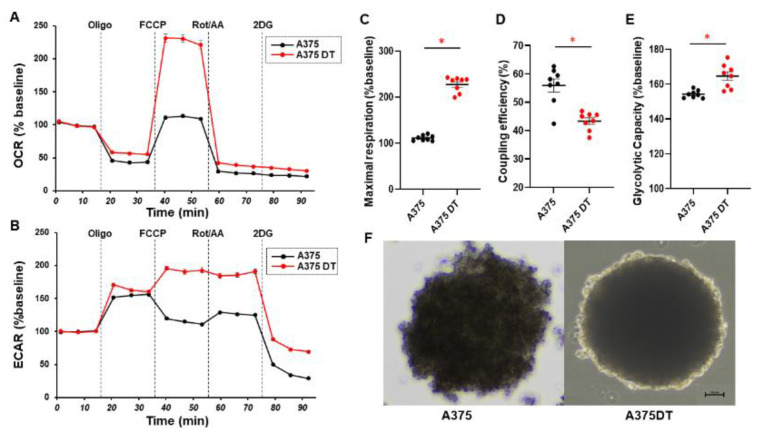
Measurement of the mitochondrial and glycolytic functions of 2D cultured A375 and A375DT cells and representative PC images of their 3D spheroids. The mitochondrial and glycolysis function of the five different 2D cultured A375 and A375DT were measured using a Seahorse XFe96 Bioanalyzer. The oxygen consumption rate (OCR) and extracellular acidification rate (ECAR) before drug injection (at the baseline) were determined as 100%. The fluctuations were then sequentially monitored by the following injections: (i) oligomycin (a complex V inhibitor), (ii) FCCP (a protonphore), (iii) rotenone/antimycin (complex I/III inhibitors), and (iv) 2-DG (a hexokinase inhibitor) (OCR; panel **A** and ECAR; panel **B**). A375DT cells showed higher maximal respiration (panel **C**), lower coupling efficiency (panel **D**), and higher glycolytic capacity (panel **E**) compared with A375 cells. Indices of mitochondrial respiration and glycolytic capacity are expressed as % baseline. * *p* < 0.05. Representative PC images of the 3D spheroids obtained from A375 and A375DT panel **F**). Scale bar; 100 μm.

**Table 1 cells-12-00759-t001:** List of the top subcategories of cancer and cellular movement.

Categories	Diseases or Functions Annotation	*p*-Value	Activations z-Score
Cancer, Organismal Injury and Abnormalities	Melanoma	6.25 × 10^−118^	2.669
Cancer, Organismal Injury and Abnormalities	Tumorigenesis of reproductive tract	2.00 × 10^−89^	2.574
Cancer, Organismal Injury and Abnormalities	Female genital neoplasm	1.05 × 10^−88^	2.574
Cancer, Gastrointestinal Disease	Upper gastrointestinal tract cancer	1.05 × 10^−63^	2.425
Cancer, Endocrine System Disorders	Gonadal tumor	2.76 × 10^−38^	2.733
Cancer, Endocrine System Disorders	Ovarian tumor	4.69 × 10^−34^	2.703
Cancer, Organismal Injury and Abnormalities	Non-hematological solid tumor	4.67 × 10^−288^	0.135
Cancer, Organismal Injury and Abnormalities	Non-hematologic malignant neoplasm	1.31 × 10^−286^	−0.425
Cancer, Organismal Injury and Abnormalities	Non-melanoma solid tumor	1.12 × 10^−283^	−0.257
Cancer, Organismal Injury and Abnormalities	Tumorigenesis of tissue	9.79 × 10^−279^	−1.779
Cancer, Organismal Injury and Abnormalities	Carcinoma	1.06 × 10^−278^	−1.242
Cancer, Organismal Injury and Abnormalities	Epithelial neoplasm	6.80 × 10^−278^	−1.482
Cancer, Organismal Injury and Abnormalities	Cancer	6.13 × 10^−277^	1.348
Cancer, Organismal Injury and Abnormalities	Malignant solid tumor	1.06 × 10^−276^	0.222
Cancer, Organismal Injury and Abnormalities	Solid tumor	2.01 × 10^−276^	1.229

**Table 2 cells-12-00759-t002:** Up-stream regulator.

Up-Stream Regulator	Expr Log Ratio	Molecule Type	Activation z-Score
IL1B	↑ 8668	cytokine	6.412
KRAS	↑ 1.404	enzyme	1.277
FGF2	↑ 1.127	growth factor	1.731
JUN	↑ 2.294	transcription regulator	1.925
EGFR	↑ 5.897	kinase	0.324
SOX2	↑ 1.519	transcription regulator	−0.283
AGT	↓ −4.28	growth factor	2.246
ESR2	↓ −7.881	ligand-dependent nuclear receptor	1.231
EGF	↓ −1.523	growth factor	2.162

**Table 3 cells-12-00759-t003:** Master regulator of causal networks.

Master Regulator	Expr Log Ratio	Molecule Type	Activations z-Score
KLP9	1.444	transcription regulator	1.477
KRAS	1.404	enzyme	1.27
SOX2	1.519	transcription regulator	2.01
TP63	8.374	transcription regulator	0.371
SMYD3	−1.272	enzyme	−0.614

**Table 4 cells-12-00759-t004:** Comparison of the mRNA expression of oncogenic signaling related factors (KRAS, SOX2, MITF, BRAF, FOS, and STAT3), ECMs (COL4, COL6, FN, and αSMA), tight-junction related molecule (ZO-1) among 2D and 3D cultured 5 MM cell lines.

		WM266-4	SM2-1	A375	MM418	SK-mel-24
KRAS	2D	cont	(↑)	→	(↑)	(↑)
	3D	cont	↑↑	↑	↑↑	↑↑
SOX2	2D	cont	↓↓	↓↓	↓↓	↓↓
	3D	cont	↓↓	→	↓↓	↑↑
STAT3	2D	cont	(↓)	(↓)	→	↑
	3D	cont	↓↓	→	↑↑	↑↑
BRAF	2D	cont	→	→	(↑)	(↑)
	3D	cont	(↓)	→	↑	↑
FOS	2D	cont	→	→	↑↑	→
	3D	cont	→	↓↓	→	↓↓
MITF	2D	cont	→	↓↓	↓↓	↓↓
	3D	cont	→	↓↓	→	↓↓
PCG1a	2D	cont	↓↓	↓↓	↓↓	↓↓
	3D	cont	↓↓	↓↓	↓↓	↓↓
COL4	2D	cont	↓↓	↓↓	↓↓	↑↑
	3D	cont	↓↓	↓↓	↓↓	↓↓
COL6	2D	cont	↓↓	↓↓	↓↓	↓↓
	3D	cont	↓↓	→	↓↓	↓↓
FN	2D	cont	↓↓	↓	↑	↓↓
	3D	cont	→	↓↓	↓↓	↓↓
aSMA	2D	cont	↓↓	↓↓	↓↓	↓
	3D	cont	↓↓	↓↓	→	↓
ZO1	2D	cont	↓↓	↓↓	↓↓	↓↓
	3D	cont	↓↓	→	↓↓	↓

no significant change; →. upregulated; (↑) = relatively, ↑ < 0.05, ↑↑ < 0.01. downregulated; (↓) = relatively, ↓ < 0.05, ↓↓ < 0.01.

**Table 5 cells-12-00759-t005:** Comparison of the mRNA expressions of factors related to oncogenic signaling (KRAS, SOX2, MITF, BRAF, FOS, and STAT3), ECMs (COL4, COL6, FN, and αSMA), tight-junction related molecule (ZO-1) between 2D and 3D cultured A375 and A375DT.

		A375	A375DT
KRAS	2D	cont	↑↑
	3D	cont	↑↑
SOX2	2D	cont	↑↑
	3D	cont	↑↑
STAT3	2D	cont	↑↑
	3D	cont	↑↑
BRAF	2D	cont	(↑)
	3D	cont	↑
FOS	2D	cont	↑↑
	3D	cont	↑↑
MITF	2D	cont	↑↑
	3D	cont	↑↑
PCG1a	2D	cont	↑↑
	3D	cont	↓↓
COL4	2D	cont	(↑)
	3D	cont	↓↓
COL6	2D	cont	↑↑
	3D	cont	↑↑
FN	2D	cont	↓↓
	3D	cont	↑↑
αSMA	2D	cont	↓↓
	3D	cont	↓↓
ZO1	2D	cont	↑↑
	3D	cont	↑↑

upregulated; (↑) = relatively, ↑ < 0.05, ↑↑ < 0.01. downregulated; ↓↓ < 0.01.

## Data Availability

The data that support the findings of this study are available from the corresponding author upon reasonable request.

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
