# Peer review of "3D Spheroid Configurations Are Possible Indictors for Evaluating the Pathophysiology of Melanoma Cell Lines"

_cells, 2023, doi:10.3390/cells12050759_

Round 1
Reviewer 1 Report (New Reviewer)
The paper by Ohguro et al needs of a significant improvement before publication. Cells is a journal with an high IF and the papers published are more complete. Here the authors organized results of preliminary evidence regarding spheroid organization linked with metabolism and gene expression. Some results are not novel (the metabolic reprograming in A375 sensitive and resistant), and moreover the changes in gene expression and in metabolism in cells 2D did not causally represent the different shape of the organoids. The authors should overexpress and/or silence some oncogenes/metabolic enzymes to have a formal validation of their involvement in the 3D structure and shape. In addition, the authors should analyze gene expression and metabolism in 3D spheroid. I hope that the authors will work on these points and then reorganize a more complete paper ready for a new submission.
Author Response
Review 1
The paper by Ohguro et al needs of a significant improvement before publication. Cells is a journal with an high IF and the papers published are more complete. Here the authors organized results of preliminary evidence regarding spheroid organization linked with metabolism and gene expression. Some results are not novel (the metabolic reprograming in A375 sensitive and resistant), and moreover the changes in gene expression and in metabolism in cells 2D did not causally represent the different shape of the organoids. The authors should overexpress and/or silence some oncogenes/metabolic enzymes to have a formal validation of their involvement in the 3D structure and shape. In addition, the authors should analyze gene expression and metabolism in 3D spheroid. I hope that the authors will work on these points and then reorganize a more complete paper ready for a new submission.
Answer; Thank you for this constructive and encouraging comment. As suggested, we performed siRNA knockdown of KRAS and/or SOX2 of most eccentric cell line, SK-mel-24. Surprisingly, these significantly reduced their horizontally deformed configuration as shown in the new Fig. 5, and also markedly altered mitochondrial and glycolytic functions. Therefore, we feel that the addition of this new added siRNA data further strengthens the whole story of the current investigation and thus this information is included in the 5th paragraph of the Result; “To elucidate what regulatory mechanisms are involved within the diversity between 3D architectures of both MM cell lines, a GO analysis, IPA up-stream and causal network analyses were performed. The results of the GO analysis based on up-regulated and down-regulated DEG suggest that several cellular mechanisms may be related, as shown in Fig. S1. Furthermore, as shown in Tables 2 and 3, 9 and 5 candidate genes as possible up-stream regulators and master regulators for the causal networks were detected, and quite interestingly, among these, KRAS and SOX2 (Fig. S2) were identified as common factors, suggesting that both factors may be critical master factors that are involved in regulating the diversity of the 3D architectures among these 5 MM cell lines. In fact, it was recently reported that SOX2 and KRAS have interconnections [39, 40], and that KRAS is mutated in 1.3% of skin cancers [39]. To verify this possibility, we next examined whether siRNA-induced knockdown of KRAS and/or SOX2 would impact the cellular metabolic functions of SK-mel-24 cell, and their 3D spheroid shape which is the most deformed among 5 MM cells. As shown in Fig. 5A, real time cellular metabolic analyses demonstrated that KRAS knockdown significantly suppressed abnormally enhanced glycolytic capacity of SK-mel-24 as well as mitochondrial maximal respiration and coupling efficiency. In contrast, the effect of SOX2 knockdown on metabolic properties was only pronounced in mitochondrial maximal respiration. Such effects were similarly observed with two different siRNAs for KRAS and SOX2 (Fig. S4). Interestingly, the knockdown of either KRAS or SOX2 significantly reduced horizontal deformation and increased the size of 3D SK-mel-24 spheroid, while the double-knockdown of KRAS and SOX2 did not induce the enlargement of the 3D spheroid. Taken together, these collective results indicate that KRAS and SOX2 were at least involved in the deformation of the 3D SK-mel-24 spheroids, but that their roles for metabolic functions were only partial. Therefore, additional factors other than KRAS and SOX2 appear to be involved in the difference in the morphology and the cellular metabolic functions among 5 MM cell lines.”.
Reviewer 2 Report (Previous Reviewer 1)
All concerns have been addressed-ready for acceptance.
Author Response
Dear Editor,
Thank you very much for the constructive comments concerning our manuscript, " 3D spheroid configurations are possible indictors for evaluating the pathophysiology of melanoma cell lines”. We examined the Editorial Decision and Reviewer's comments carefully and prepared a revised version of our paper that takes these comments into account for resubmission. Therefore, we will greatly appreciate it if you will consider our revised paper for possible publication in Cells. The changes are listed below.
Editorial Decision
So far, 4 out of 6 reviewers have recommended to accept the paper with minor or major revisions, while 2 reviewers have recommended to reject due to their concern that the results do not support the conclusions, and require additional experiments such as overexpression and/or silencing of KRAS and SOX2 to validate their role as critical regulators of the 3D structure and shape. The authors have included the lack of these experiments as simply a limitation (in the Discussion section). I agree with the 4 out of 6 reviewers (the majority) and have decided to accept (pending major revision). However, I also agree that the 2 other reviewer's major concern is valid and, thus, to support the authors conclusions that KRAS and SOX2 are "critical regulators" of spatial architecture of melanoma cells lines (as stated in their title), it is necessary to include at the very least one experiment with siRNA knockdown showing that knockdown of KRAS and or SOX2 expression can impact the horizontal and vertical eccentricities of 3D spheroids, and spatial architecture of the MM cells.
Answer; Thank you this constructive and encouraging comment. As suggested, we performed the siRNA knockdown of KRAS and/or SOX2 of most eccentric cell line, SK-mel-24. Surprisingly, these significantly reduced their horizontal deformed configuration as shown in the new Fig. 5, and also markedly altered mitochondrial and glycolytic functions. Therefore, these findings are included in the 5th paragraph of the Result; “To elucidate what regulatory mechanisms are involved within the diversity between 3D architectures of both MM cell lines, a GO analysis, IPA up-stream and causal network analyses were performed. The results of the GO analysis based on up-regulated and down-regulated DEG suggest that several cellular mechanisms may be related, as shown in Fig. S1. Furthermore, as shown in Tables 2 and 3, 9 and 5 candidate genes as possible up-stream regulators and master regulators for the causal networks were detected, and quite interestingly, among these, KRAS and SOX2 (Fig. S2) were identified as common factors, suggesting that both factors may be critical master factors that are involved in regulating the diversity of the 3D architectures among these 5 MM cell lines. In fact, it was recently reported that SOX2 and KRAS have interconnections [39, 40], and that KRAS is mutated in 1.3% of skin cancers [39]. To verify this possibility, we next examined whether siRNA-induced knockdown of KRAS and/or SOX2 would impact the cellular metabolic functions of SK-mel-24 cell, and their 3D spheroid shape which is the most deformed among 5 MM cells. As shown in Fig. 5A, real time cellular metabolic analyses demonstrated that KRAS knockdown significantly suppressed abnormally enhanced glycolytic capacity of SK-mel-24 as well as mitochondrial maximal respiration and coupling efficiency. In contrast, the effect of SOX2 knockdown on metabolic properties was only pronounced in mitochondrial maximal respiration. Such effects were similarly observed with two different siRNAs for KRAS and SOX2 (Fig. S4). Interestingly, the knockdown of either KRAS or SOX2 significantly reduced horizontal deformation and increased the size of 3D SK-mel-24 spheroid, while the double-knockdown of KRAS and SOX2 did not induce the enlargement of the 3D spheroid. Taken together, these collective results indicate that KRAS and SOX2 were at least involved in the deformation of the 3D SK-mel-24 spheroids, but that their roles for metabolic functions were only partial. Therefore, additional factors other than KRAS and SOX2 appear to be involved in the difference in the morphology and the cellular metabolic functions among 5 MM cell lines.”.
Review 1
The paper by Ohguro et al needs of a significant improvement before publication. Cells is a journal with an high IF and the papers published are more complete. Here the authors organized results of preliminary evidence regarding spheroid organization linked with metabolism and gene expression. Some results are not novel (the metabolic reprograming in A375 sensitive and resistant), and moreover the changes in gene expression and in metabolism in cells 2D did not causally represent the different shape of the organoids. The authors should overexpress and/or silence some oncogenes/metabolic enzymes to have a formal validation of their involvement in the 3D structure and shape. In addition, the authors should analyze gene expression and metabolism in 3D spheroid. I hope that the authors will work on these points and then reorganize a more complete paper ready for a new submission.
Answer; Thank you for this constructive and encouraging comment. As suggested, we performed siRNA knockdown of KRAS and/or SOX2 of most eccentric cell line, SK-mel-24. Surprisingly, these significantly reduced their horizontally deformed configuration as shown in the new Fig. 5, and also markedly altered mitochondrial and glycolytic functions. Therefore, we feel that the addition of this new added siRNA data further strengthens the whole story of the current investigation and thus this information is included in the 5th paragraph of the Result; “To elucidate what regulatory mechanisms are involved within the diversity between 3D architectures of both MM cell lines, a GO analysis, IPA up-stream and causal network analyses were performed. The results of the GO analysis based on up-regulated and down-regulated DEG suggest that several cellular mechanisms may be related, as shown in Fig. S1. Furthermore, as shown in Tables 2 and 3, 9 and 5 candidate genes as possible up-stream regulators and master regulators for the causal networks were detected, and quite interestingly, among these, KRAS and SOX2 (Fig. S2) were identified as common factors, suggesting that both factors may be critical master factors that are involved in regulating the diversity of the 3D architectures among these 5 MM cell lines. In fact, it was recently reported that SOX2 and KRAS have interconnections [39, 40], and that KRAS is mutated in 1.3% of skin cancers [39]. To verify this possibility, we next examined whether siRNA-induced knockdown of KRAS and/or SOX2 would impact the cellular metabolic functions of SK-mel-24 cell, and their 3D spheroid shape which is the most deformed among 5 MM cells. As shown in Fig. 5A, real time cellular metabolic analyses demonstrated that KRAS knockdown significantly suppressed abnormally enhanced glycolytic capacity of SK-mel-24 as well as mitochondrial maximal respiration and coupling efficiency. In contrast, the effect of SOX2 knockdown on metabolic properties was only pronounced in mitochondrial maximal respiration. Such effects were similarly observed with two different siRNAs for KRAS and SOX2 (Fig. S4). Interestingly, the knockdown of either KRAS or SOX2 significantly reduced horizontal deformation and increased the size of 3D SK-mel-24 spheroid, while the double-knockdown of KRAS and SOX2 did not induce the enlargement of the 3D spheroid. Taken together, these collective results indicate that KRAS and SOX2 were at least involved in the deformation of the 3D SK-mel-24 spheroids, but that their roles for metabolic functions were only partial. Therefore, additional factors other than KRAS and SOX2 appear to be involved in the difference in the morphology and the cellular metabolic functions among 5 MM cell lines.”.
Reviewer 3
In my first report I have recommended publication after some minor corrections. The authors have adjusted the manuscript and although I feel it is a manuscript based on preliminary results, I continue believing it deserves to be published, once author affirm no study focusing in the underlying mechanisms responsible for inducing 3D spheroid architectures on melanoma cells, has been reported so far.
Answer; Thank you for this constructive and encouraging comment. As suggested by this reviewer as well as the Editorial stuff, we performed the siRNA knockdown of KRAS and/or SOX2 of the most unusual cell line, SK-mel-24. Surprisingly, these actions significantly reduced their horizontally deformed configurations, as shown in the new Fig. 5, and also markedly altered mitochondrial and glycolytic functions. Therefore, we feel that the addition of this new added siRNA data further strengthens the whole story of the current investigation and thus these findings are included in the 5th paragraph of the Result; “To elucidate what regulatory mechanisms are involved within the diversity between 3D architectures of both MM cell lines, a GO analysis, IPA up-stream and causal network analyses were performed. The results of the GO analysis based on up-regulated and down-regulated DEG suggest that several cellular mechanisms may be related, as shown in Fig. S1. Furthermore, as shown in Tables 2 and 3, 9 and 5 candidate genes as possible up-stream regulators and master regulators for the causal networks were detected, and quite interestingly, among these, KRAS and SOX2 (Fig. S2) were identified as common factors, suggesting that both factors may be critical master factors that are involved in regulating the diversity of the 3D architectures among these 5 MM cell lines. In fact, it was recently reported that SOX2 and KRAS have interconnections [39, 40], and that KRAS is mutated in 1.3% of skin cancers [39]. To verify this possibility, we next examined whether siRNA-induced knockdown of KRAS and/or SOX2 would impact the cellular metabolic functions of SK-mel-24 cell, and their 3D spheroid shape which is the most deformed among 5 MM cells. As shown in Fig. 5A, real time cellular metabolic analyses demonstrated that KRAS knockdown significantly suppressed abnormally enhanced glycolytic capacity of SK-mel-24 as well as mitochondrial maximal respiration and coupling efficiency. In contrast, the effect of SOX2 knockdown on metabolic properties was only pronounced in mitochondrial maximal respiration. Such effects were similarly observed with two different siRNAs for KRAS and SOX2 (Fig. S4). Interestingly, the knockdown of either KRAS or SOX2 significantly reduced horizontal deformation and increased the size of 3D SK-mel-24 spheroid, while the double-knockdown of KRAS and SOX2 did not induce the enlargement of the 3D spheroid. Taken together, these collective results indicate that KRAS and SOX2 were at least involved in the deformation of the 3D SK-mel-24 spheroids, but that their roles for metabolic functions were only partial. Therefore, additional factors other than KRAS and SOX2 appear to be involved in the difference in the morphology and the cellular metabolic functions among 5 MM cell lines.”.

Reviewer 3 Report (Previous Reviewer 3)
In my first report I have recommended publication after some minor corrections. The authors have adjusted the manuscript and although I feel it is a manuscript based on preliminary results, I continue believing it deserves to be published, once author affirm no study focusing in the underlying mechanisms responsible for inducing 3D spheroid architectures on melanoma cells, has been reported so far.
Author Response
Reviewer 3
In my first report I have recommended publication after some minor corrections. The authors have adjusted the manuscript and although I feel it is a manuscript based on preliminary results, I continue believing it deserves to be published, once author affirm no study focusing in the underlying mechanisms responsible for inducing 3D spheroid architectures on melanoma cells, has been reported so far.
Answer; Thank you for this constructive and encouraging comment. As suggested by this reviewer as well as the Editorial stuff, we performed the siRNA knockdown of KRAS and/or SOX2 of the most unusual cell line, SK-mel-24. Surprisingly, these actions significantly reduced their horizontally deformed configurations, as shown in the new Fig. 5, and also markedly altered mitochondrial and glycolytic functions. Therefore, we feel that the addition of this new added siRNA data further strengthens the whole story of the current investigation and thus these findings are included in the 5th paragraph of the Result; “To elucidate what regulatory mechanisms are involved within the diversity between 3D architectures of both MM cell lines, a GO analysis, IPA up-stream and causal network analyses were performed. The results of the GO analysis based on up-regulated and down-regulated DEG suggest that several cellular mechanisms may be related, as shown in Fig. S1. Furthermore, as shown in Tables 2 and 3, 9 and 5 candidate genes as possible up-stream regulators and master regulators for the causal networks were detected, and quite interestingly, among these, KRAS and SOX2 (Fig. S2) were identified as common factors, suggesting that both factors may be critical master factors that are involved in regulating the diversity of the 3D architectures among these 5 MM cell lines. In fact, it was recently reported that SOX2 and KRAS have interconnections [39, 40], and that KRAS is mutated in 1.3% of skin cancers [39]. To verify this possibility, we next examined whether siRNA-induced knockdown of KRAS and/or SOX2 would impact the cellular metabolic functions of SK-mel-24 cell, and their 3D spheroid shape which is the most deformed among 5 MM cells. As shown in Fig. 5A, real time cellular metabolic analyses demonstrated that KRAS knockdown significantly suppressed abnormally enhanced glycolytic capacity of SK-mel-24 as well as mitochondrial maximal respiration and coupling efficiency. In contrast, the effect of SOX2 knockdown on metabolic properties was only pronounced in mitochondrial maximal respiration. Such effects were similarly observed with two different siRNAs for KRAS and SOX2 (Fig. S4). Interestingly, the knockdown of either KRAS or SOX2 significantly reduced horizontal deformation and increased the size of 3D SK-mel-24 spheroid, while the double-knockdown of KRAS and SOX2 did not induce the enlargement of the 3D spheroid. Taken together, these collective results indicate that KRAS and SOX2 were at least involved in the deformation of the 3D SK-mel-24 spheroids, but that their roles for metabolic functions were only partial. Therefore, additional factors other than KRAS and SOX2 appear to be involved in the difference in the morphology and the cellular metabolic functions among 5 MM cell lines.”.
Round 2
Reviewer 1 Report (New Reviewer)
The paper is improved adding the novel results. Now it is acceptable for publication.
This manuscript is a resubmission of an earlier submission. The following is a list of the peer review reports and author responses from that submission.
Round 1
Reviewer 1 Report
Article by Dr. Uhara and group elaborating on the role of KRAS and SOX2 in melanoma cancer. This is a very interesting study with well-planned experiments and a well-written manuscript. Though few things need to be addressed before it is ready for acceptance. They are as follows:
1. It has been shown recently (PMID: 33870211 and PMID: 32728033) that SOX2 and KRAS have interconnection. It has also been shown that KRAS is mutated in 1.3% of skin cancers. These two pieces of information must be discussed in the introduction part with relevant mentioned references.
2. Figure 4 resolution must be rectified and improved properly.
3. Authors should add a model at the end of the manuscript depicting the theme of this manuscript and discussing it briefly.
Author Response
Reviewer 1
Article by Dr. Uhara and group elaborating on the role of KRAS and SOX2 in melanoma cancer. This is a very interesting study with well-planned experiments and a well-written manuscript. Though few things need to be addressed before it is ready for acceptance. They are as follows:
- It has been shown recently (PMID: 33870211 and PMID: 32728033) that SOX2 and KRAS have interconnection. It has also been shown that KRAS is mutated in 1.3% of skin cancers. These two pieces of information must be discussed in the introduction part with relevant mentioned references.
Answer; Thank you for this excellent advice. As suggested, this valuable information is now included in the 2nd paragraph of the Discussion section; “As possible mechanisms responsible for the differences between the 2D and 3D cultures of 3T3-L1 preadipocytes, our recently reported transcriptome analysis using an IPA upstream analysis identified STAT3 as the master upstream gene as the regulator responsible for inducing the diverse biological properties between these culture systems [38]. Among the STAT family proteins, STAT1 to STAT6 [39], it was revealed that STAT3 functionally plays an important role in modulating the biological activities of cancer cells, by affecting their energy metabolism, and the metabolism of glucose and lipids [40, 41]. Furthermore, since STAT3 is also involved in gravity-induced biological activities [42, 43], these observations strongly suggest that such biological difference between 2D and 3D culture systems are caused by STAT3 related signaling mechanisms, suggesting that similar mechanisms may also be involved in the diversity observed within the 3D spheroid configurations of MM cell lines. In fact, in the current study, the gene expression of STAT3 as well as BRAF indeed fluctuated to reflect the 3D architectures of the 3D spheroids among 5 MM cells. Interestingly, findings reported in recent studies suggest that STAT3 signaling is closely linked with KRAS and SOX2 related networks as follows; 1) STAT3 regulates the epithelial differentiation of malignant tumors caused by oncogenic KRAS, and thus, STAT3 acts as a regulator for cellular plasticity and the inhibition of the epithelial mesenchymal transition (EMT) linked with metastasis [44], 2) STAT3 can be upregulated in cancer stem cells [45] together with SOX2 in clustered circulating tumor cells, resulting in a higher potential for developing metastasis [46], and 3) a BRAF inhibitor initiates the STAT3 activation caused the up-regulation of SOX2 and CD24 resulting an increased tolerance against BRAF inhibitors [47, 48]. Therefore, these collective evidences strongly suggest that STAT3 related SOX2 and KRAS networks may be involved in the tumor metastasis and drug resistance in MM cells, and caused us to speculate that 3D spheroid configurations of MM cells may be valuable indicators for estimating their pathological activities. Since it has been reported that only 2 % of skin MM are associated with KRAS mutations [51, 52] but that SOX2 is expressed in 45 % of primary MMs and a 40 % MM metastasis [53], we rationally speculated that wild-type KRAS and SOX2 may be involved in this mechanism. In fact, to effectively target oncogenic RAS including KRAS with their downstream signaling and metabolic pathways during tumorigenesis, we examined those that were connected with several other oncogenic driver genes including SOX2 [54, 55]. Furthermore, and quite interestingly, it was shown that SOX2 modulates the levels of MITF, a key determinant of the MM phenotype [56], in human melanocytes, and MM lines in vitro [57].”
- Figure 4 resolution must be rectified and improved properly.
Answer; Thank you for this comment. We agree that this figure may not be sharp enough. Nevertheless, this figure was down-loaded from the IPA analysis data, and therefore, this was best resolution that could be obtained. However, in the current study, this graphic summary may not be critical. Therefore, this figure was omitted.
- Authors should add a model at the end of the manuscript depicting the theme of this manuscript and discussing it briefly.
Answer; Thank you for this comment. As suggested, a possible model was included in the new Fig. 6.
Reviewer 2 Report
Study presented by Ohguro et al. revealed that the 3D organisation of melanoma cell lines is associated with their metabolic status and transcriptional phenotype. They identified KRAS and SOX2 as crucial regulators of spheroid architecture.
While this is an interesting approach to look at interpatient heterogeneity in melanoma, the authors need to perform additional experiments to support their conclusions, especially about the role of KRAS and SOX2.
1) Could authors stain for KRAS and SOX2in addition to MITF in melanospheres? Studies from Haass lab shown that the heterogeneity of melanoma spheroid is dictated by MITF, it would be interesting to see whether there is a correlation between KRAS/SOX2 and MITF.
2) Could authors define better the deformity of the spheroids and the way they categorized spheroids according to their shape? There is a way to quantify this observation (e.g. eccentricity measurement)?
3) Role of metabolism in spheroid organization: could authors check if the manipulation of glycolytic and mitochondrial metabolism (e. g pharmacological inhibition of glycolysis or mitochondrial respiration) can change the deformity of the spheres?
4) Role of KRAS and SOX2 in melanoma architecture: could authors affect spheroid architecture and metabolism by depleting and/or overexpressing KRAS and SOX2 in MM cell lines? This experiments would support the statement of the crucial role of KRAS and SOX2 in spheroid architecture.
5) There is the correlation of melanospheres architecture and response to treatment? In the introduction, the authors described the 3D cultures as the alternative approach to model disease progression/response to therapy. Could authors show if it is possible to predict the response to treatment (immune checkpoint inhibitors/ targeted therapies) based on melanosphere architecture? Is this KRAS and SOX2-dependent?
Other comments:
Figure 3: The legend is wrong and you can the cropping is not precise (you can see half of the lab above the panel A)
Author Response
Study presented by Ohguro et al. revealed that the 3D organisation of melanoma cell lines is associated with their metabolic status and transcriptional phenotype. They identified KRAS and SOX2 as crucial regulators of spheroid architecture.
While this is an interesting approach to look at interpatient heterogeneity in melanoma, the authors need to perform additional experiments to support their conclusions, especially about the role of KRAS and SOX2.
- Could authors stain for KRAS and SOX2in addition to MITF in melanospheres? Studies from Haass lab shown that the heterogeneity of melanoma spheroid is dictated by MITF, it would be interesting to see whether there is a correlation between KRAS/SOX2 and MITF.
Answer; Thank you so much for this excellent proposal. As suggested, this information is now included within the 2nd paragraph of Introduction; “During the development of translation research related to MM, several diseases models have been used to obtain fundamental discoveries in MM biology and these studies have resulted in the development of new therapeutic targets and the identification of several clinical markers as well as others. For example, several studies using animal models have revealed the efficacy of using anti-PD-1 and anti-CTLA-4 antibodies to inhibit tumor immunity [11-13], leading to therapy for patients with MM and other malignancies [14, 15]. As an alternative approach, in vitro three-dimensional (3D) culture models, which have been used in studies concerning the structure, molecular and cellular mechanisms of tumors as compared to the conventional two-dimensional (2D) models, have also been utilized in research related to malignant tumors including MM [16]. In fact, several studies reported that these in vitro 3D tumor models facilitate a better understanding of cell to cell and cell to matrix interactions since they replicate the architecture and cellular heterogeneity of tumor tissues more closely [17, 18]. Thus, consequently, these in vitro 3D models are now frequently used for the screening of new antitumor drugs [19, 20]. Among the several in vitro 3D models, 3D spheroids, which are simpler forms and can be produced in a nonadherent surface manner, have been the most widely used [21, 22], and in fact, they have emerged as a useful tool for modeling a number of human diseases, in addition to malignant tumors including MM [23-25]. Interestingly, using a 3D MM spheroid model, Ahmed and Haass reported that the proliferative and invasive efficacies could be defined by the high and low expression, respectively, of the microphthalmia-associated transcription factor (MITF) [26].”, and 2nd paragraph of Discussion; “As possible mechanisms responsible for the differences between the 2D and 3D cultures of 3T3-L1 preadipocytes, our recently reported transcriptome analysis using an IPA upstream analysis identified STAT3 as the master upstream gene as the regulator responsible for inducing the diverse biological properties between these culture systems [40]. Among the STAT family proteins, STAT1 to STAT6 [41], it was revealed that STAT3 functionally plays an important role in modulating the biological activities of cancer cells, by affecting their energy metabolism, and the metabolism of glucose and lipids [42, 43]. Furthermore, since STAT3 is also involved in gravity-induced biological activities [44, 45], these observations strongly suggest that such biological difference between 2D and 3D culture systems are caused by STAT3 related signaling mechanisms, suggesting that similar mechanisms may also be involved in the diversity observed within the 3D spheroid configurations of MM cell lines. In fact, in the current study, the gene expression of STAT3 as well as BRAF indeed fluctuated to reflect the 3D architectures of the 3D spheroids among 5 MM cells. Interestingly, findings reported in recent studies suggest that STAT3 signaling is closely linked with KRAS and SOX2 related networks as follows; 1) STAT3 regulates the epithelial differentiation of malignant tumors caused by oncogenic KRAS, and thus, STAT3 acts as a regulator for cellular plasticity and the inhibition of the epithelial mesenchymal transition (EMT) linked with metastasis [46], 2) STAT3 can be upregulated in cancer stem cells [47] together with SOX2 in clustered circulating tumor cells, resulting in a higher potential for developing metastasis [48], and 3) a BRAF inhibitor initiates the STAT3 activation caused the up-regulation of SOX2 and CD24 resulting an increased tolerance against BRAF inhibitors [49, 50]. Therefore, these collective evidences strongly suggest that STAT3 related SOX2 and KRAS networks may be involved in the tumor metastasis and drug resistance in MM cells, and caused us to speculate that 3D spheroid configurations of MM cells may be valuable indicators for estimating their pathological activities. Since it has been reported that only 2 % of skin MM are associated with KRAS mutations [51, 52] but that SOX2 is expressed in 45 % of primary MMs and a 40 % MM metastasis [53], we rationally speculated that wild-type KRAS and SOX2 may be involved in this mechanism. In fact, to effectively target oncogenic RAS including KRAS with their downstream signaling and metabolic pathways during tumorigenesis, we examined those that were connected with several other oncogenic driver genes including SOX2 [54, 55]. Furthermore, and quite interestingly, it was shown that SOX2 modulates the levels of MITF, a key determinant of the MM phenotype [56], in human melanocytes, and MM lines in vitro [57].”
- Could authors define better the deformity of the spheroids and the way they categorized spheroids according to their shape? There is a way to quantify this observation (e.g. eccentricity measurement)?
Answer; Thank you for this comment. In terms of the degree of deformity of the 3D MM spheroids, we determined the ratios of the outer circle to the inner circle using the downward view PC image of the 3D spheroid as the eccentricity measurement as shown in the new Fig. 2.
- Role of metabolism in spheroid organization: could authors check if the manipulation of glycolytic and mitochondrial metabolism (e. g pharmacological inhibition of glycolysis or mitochondrial respiration) can change the deformity of the spheres?
- Role of KRAS and SOX2 in melanoma architecture: could authors affect spheroid architecture and metabolism by depleting and/or overexpressing KRAS and SOX2 in MM cell lines? This experiment would support the statement of the crucial role of KRAS and SOX2 in spheroid architecture.
Answer;
- There is the correlation of melanospheres architecture and response to treatment? In the introduction, the authors described the 3D cultures as the alternative approach to model disease progression/response to therapy. Could authors show if it is possible to predict the response to treatment (immune checkpoint inhibitors/ targeted therapies) based on melanosphere architecture? Is this KRAS and SOX2-dependent?
Answers for #3-5; Thank you so much for these excellent experimental proposals. These suggested points are quite interesting and will be required to develop a better understanding of our current observations. Therefore, these issues are included in the study limitations within the Discussion section of the paper; “However, as of this writing, this conclusion remains speculative and the following study limitations would need to be investigated regarding these interesting and unidentified issues; 1) the relationship between cellular metabolism and 3D spheroid organization, 2) the roles of KRAS and SOX2 in the melanoma architecture, and 3) the correlation of the 3D spheroid architecture and the response to treatment because of the differences in the expression of BRAF among 5 MM cell lines. Thus, to understand the possible biological correlations between 3D spheroid configurations and the tumor pathogenesis of MM, we plan to investigate following issues; studies of 1) spheroid architecture and metabolism by depleting and/or overexpressing KRAS and SOX2, 2) the deformity of the 3D spheroids by manipulating glycolytic and mitochondrial metabolism, e. g. the pharmacological inhibition of glycolysis or mitochondrial respiration, 3) the response to various treatments including immune checkpoint inhibitors/ targeted therapies based on differences in the 3D spheroid architecture, and others.”
Other comments:
- Figure 3: The legend is wrong and you can the cropping is not precise (you can see half of the lab above the panel A)
Answer; Thank you for this comment. As you pointed out, the corresponding legend (changed to Fig. 4 due to inclusion of new Fig. 2) was changed; “DEGs (cutoff false discovery rate (FDR) < 0.05 and/or the magnitude of change ≥2) between 2D cultured WM266-4 and SK-mel-24 cells were demonstrated by hierarchical clustering heatmaps (panel A), M-A plots (panel B) and volcano plots (panel C). Colored bars and points represent DEGs that are either over-expressed (red) or under-expressed (blue) in WM266-4 with SK-mel-24. “.
Reviewer 3 Report
The manuscript aimed to elucidate the molecular mechanisms responsible for the spatial proliferation of malignant melanomas (MM). For this purpose, three-dimension (3D) spheroids were produced from five MM cell lines and their 3D architectures and mitochondrial and glycolytic metabolism were evaluated.
The group demonstrated knowledge on this area, as recently they have succeeded in producing 3D spheroids of several non-cancerous cells. The group assumes that no study focusing the molecular mechanisms responsible for causing 3D spheroid architectures (globe shaped or non-globe shaped), has previously been reported.
Based on results, the group speculate that the 3D spheroid configuration may be related to the pathophysiological activities of MM. The manuscript is well written and desserves to be published quickly.
As minor correction, I point out the below:
Line 23: the most deformed ones
Line 53: “as well as others” ... please, specify.
Please, remove the word “various” from the cell lines sentences:
Line 92: “Five various human MM cell lines including..”
Line 113: “of the various 2D”
Line 158: various
Author Response
"The manuscript aimed to elucidate the molecular mechanisms responsible for the spatial proliferation of malignant melanomas (MM). For this purpose, three-dimension (3D) spheroids were produced from five MM cell lines and their 3D architectures and mitochondrial and glycolytic metabolism were evaluated.
The group demonstrated knowledge on this area, as recently they have succeeded in producing 3D spheroids of several non-cancerous cells. The group assumes that no study focusing the molecular mechanisms responsible for causing 3D spheroid architectures (globe shaped or non-globe shaped), has previously been reported.
Based on results, the group speculate that the 3D spheroid configuration may be related to the pathophysiological activities of MM. The manuscript is well written and desserves to be published quickly.
As minor correction, I point out the below:
- Line 23: the most deformed ones
Answer; Thank you for this comment. As pointed out, that was change to “the most deformed ones”.
- Line 53: “as well as others” ... please, specify.
Answer; Thank you for this comment. As pointed out, that was ambiguous and may cause misunderstanding. Therefore, those “as well as others” were removed..
Please, remove the word “various” from the cell lines sentences:
Line 92: “Five various human MM cell lines including..”
Answer; Thank you for this comment. As suggested, that was removed.
- Line 113: “of the various 2D”
Answer; Thank you for this comment. As suggested, that was removed.
- Line 158: various"
Answer; Thank you for this comment. As suggested, that was removed.
Round 2
Reviewer 1 Report
Authors must update the references, it's not matching with the text. New references are not there in the revised manuscript.
Reviewer 2 Report
I would like to thank the authors for adjusting the text according to reviewer's comments. Unfortunately, author’s didn’t add any experimental data to support their conclusions regarding the role of SOX2 and KRAS in the spatial organization of melanospheres. Therefore, I don’t think that the current version of the manuscript is suitable for publication in MDPI Cells.